# Ulcerative Colitis and Acute Severe Ulcerative Colitis Patients Are Overlooked in Infliximab Population Pharmacokinetic Models: Results from a Comprehensive Review

**DOI:** 10.3390/pharmaceutics14102095

**Published:** 2022-09-30

**Authors:** Alix Démaris, Ella S. K. Widigson, Johan F. K. F. Ilvemark, Casper Steenholdt, Jakob B. Seidelin, Wilhelm Huisinga, Robin Michelet, Linda B. S. Aulin, Charlotte Kloft

**Affiliations:** 1Department of Clinical Pharmacy and Biochemistry, Institute of Pharmacy, Freie Universität Berlin, Kelchstrasse 31, 12169 Berlin, Germany; 2Graduate Research Training Program PharMetrX, 12169 Berlin, Germany; 3Department of Gastroenterology, Copenhagen University Hospital Herlev, 2730 Herlev, Denmark; 4Institute of Mathematics, Universität Potsdam, 14476 Potsdam, Germany

**Keywords:** infliximab, inflammatory bowel disease, ulcerative colitis, acute severe ulcerative colitis, ulcerative colitis, disease activity, pharmacokinetic, pharmacometrics

## Abstract

Ulcerative colitis (UC) is part of the inflammatory bowels diseases, and moderate to severe UC patients can be treated with anti-tumour necrosis α monoclonal antibodies, including infliximab (IFX). Even though treatment of UC patients by IFX has been in place for over a decade, many gaps in modelling of IFX PK in this population remain. This is even more true for acute severe UC (ASUC) patients for which early prediction of IFX pharmacokinetic (PK) could highly improve treatment outcome. Thus, this review aims to compile and analyse published population PK models of IFX in UC and ASUC patients, and to assess the current knowledge on disease activity impact on IFX PK. For this, a semi-systematic literature search was conducted, from which 26 publications including a population PK model analysis of UC patients receiving IFX therapy were selected. Amongst those, only four developed a model specifically for UC patients, and only three populations included severe UC patients. Investigations of disease activity impact on PK were reported in only 4 of the 14 models selected. In addition, the lack of reported model codes and assessment of predictive performance make the use of published models in a clinical setting challenging. Thus, more comprehensive investigation of PK in UC and ASUC is needed as well as more adequate reports on developed models and their evaluation in order to apply them in a clinical setting.

## 1. Introduction

Inflammatory bowel diseases (IBD) are a group of chronic, inflammatory disorders affecting the gastrointestinal (GI) tract, with the two main forms being ulcerative colitis (UC) and Crohn’s disease (CD). The aetiology is not fully known, but both genetic and environmental factors play an important role in the development of IBD [1]. The environmental component of disease development can be observed by the much higher prevalence of IBD in northern Europe and North America than the rest of the world, with over 200 cases per 100,000 inhabitants with a somewhat higher prevalence of UC than CD [2]. Even though their incidences are comparable, there are noteworthy differences between the two conditions’ disease characteristics.

UC affects only the mucosal and submucosal layers of the colon and rectum, while inflammation in CD can occur transmurally anywhere across the digestive tract [2]. The lesions are also distributed differently, with the inflammation extending in a continuous retrograde mode in UC, but with unaffected sections of the GI between inflamed areas in CD. In both conditions, however, disease activity has been associated with elevated levels of the pro-inflammatory cytokine tumour necrosis factor alpha (TNFα), which is a key player in the initiation and modulation of an inflammatory reaction [3]. Due to the large differences in disease phenotype, it is crucial to evaluate the efficacy of pharmacological interventions for the specific IBD patient populations. As it was the authors’ impression that UC patients were underrepresented in different PK studies of IBD patients, this review focuses solely on UC patients.

Multiple indices exist for assessing UC disease activity, although the most commonly used in clinical trials is the Mayo Score, either as the full Mayo Score (fMayo), partial Mayo Score (pMayo) or Mayo Endoscopic Score (MES) [4,5]:pMayo comprises solely non-endoscopic measures (stool frequency, rectal bleeding and global medical assessment);MES instead comprises solely endoscopic findings;fMayo is the combination of the indices pMayo and MES [6].

The Simple Clinical Colitis Activity Index (SCCAI) can also be used for disease assessment of UC patients. Similarly to the pMayo, it is solely based on symptom manifestations that the patient can report on themselves (stool frequency during day, stool frequency during night, urgency of defecation, blood in stool, general well-being and extracolonic features) [7]. While most UC patients experience a mild to moderate course of disease, with interchanging periods of activity and remission, 15–25% of patients will at some point experience a flare of acute severe UC (ASUC) [8,9,10]. ASUC is defined as a high clinical score of disease severity in combination with extensive and deep ulcerations seen on endoscopy, increasing the risk of bowel wall perforation and the need for acute colectomy [11].

Acute colectomy has a significantly higher mortality rate compared to elective surgery (odds ratio, OR 1.82; confidence interval, CI 95% 1.19–2.63) [12]. Consequently, patients with ASUC are usually intensively monitored at the hospital and receive an initial treatment of intravenous corticosteroids [11,13,14]. Although corticosteroid treatment was shown in an early clinical trial to decrease the mortality in UC patients from 24% to 7%, 30–50% of ASUC patients fail to respond to the initial treatment and are then switched to treatment with infliximab (IFX), in the case that acute colectomy is not imminently needed [13,14,15].

IFX is not only used in the treatment of ASUC, but also more widely used in the management of moderate to severe UC and CD, together with other various immune mediated inflammatory diseases [16,17]. IFX is a chimeric immunoglobulin G1 monoclonal antibody (mAb), asserting its effect by targeting TNFα (Appendix A) and was the first mAb approved for the treatment of IBD [18,19]. As TNFα is a key player in driving the mucosal immune response in UC, blocking TNFα with IFX promotes mucosal healing and clinical remission [19]. IFX can be used to remove corticosteroids therapy or to replace it when patients become non-responsive. Other treatments might also be used to treat UC patients after failure to corticosteroids such as thiopurines, other anti-TNFα biologics (adalimumab, golimumab), vedolizumab, ustekinumab or tofacitinib [3,20,21]. However, UC and CD patients refractory to corticosteroids are often treated with biologics, amongst which IFX is the most common [21,22]. IFX is given intravenously, with a standard dosing regimen comprising an induction phase with 5 mg/kg administered at week 0, 2 and 6, followed by a maintenance phase with administrations every 8 weeks [23]. Rescue therapy with IFX in corticosteroid-refractory ASUC has been shown to reduce the number of patients needing colectomy during hospitalization by half [24]. However, there is still a 10% risk of acute colectomy amongst these patients, and others may lose response over time, warranting further improvements of drug therapy and strategy [24].

The utility of therapeutic drug monitoring (TDM), in which serum concentrations collected at trough (or minimum concentration, C_min_) are compared to empirically established concentration thresholds, has been widely acknowledged as a tool to optimise drug therapy with biologics for IBD [25]. TDM can be performed as a response to observed treatment failure in the individual patient (i.e., reactive TDM), or routinely to be able to detect insufficient exposure before patients fail therapy (i.e., proactive TDM). A relationship has been established between treatment efficacy and IFX exposure, with IFX C_min_ ≥ 5 µg/mL during maintenance therapy being linked to clinical remission [26]. However, the variability in IFX pharmacokinetics (PK) between patients, and increased inflammatory load, assessed by increased serum concentration of C-reactive protein (CRP), decreased albumin concentration, and intestinal loss of IFX via faeces, has been related to correlate with increased IFX clearance [19]. Since ASUC is caused by increased bowel inflammation, patients with ASUC characterised by very severe inflammation would hypothetically be prone to treatment failure due to insufficient IFX exposure.

Another factor known to cause decreased IFX exposure and subsequent risk of treatment failure is the production of anti-drug antibodies (ADA) [27]. A meta-analysis conducted in 2013 reported that one in three IBD patients developing ADA also experienced a loss of clinical response (LOR) to IFX treatment during maintenance therapy [28]. In addition, approximately one-third of patients show no initial response to IFX induction therapy, referred to as primary non-responders [29]. Thus, it is clear that IFX therapy, in particular its dosing strategy, needs optimization.

To further individualise and optimise IFX treatment, it is important to understand and quantify how patient-, drug-, and disease-related factors such as ADA production and inflammatory burden influence IFX PK and pharmacodynamics (PD). Knowledge about both disease and the drug PK can be combined into pharmacometric models. Leveraging a pharmacometric model, together with individual influential factors (covariates) and the individual’s measured circulating drug concentrations, to predict an appropriate dosing regimen, is termed model-informed precision dosing (MIPD) [30,31]. The use of MIPD allows for more precise individual dosing adjustments compared to conventional TDM, and thus has the potential to further individualise and improve treatments [32]. Notably, a cornerstone for successful MIPD is the ability of the PK model to adequately characterise and predict the PK for the relevant (sub)populations as well as for each individual patient. Hence, for a certain PK model to be utilised for MIPD purposes, this model’s performance needs to be assessed by adequate model evaluation, and a published model needs to be well documented to allow for (re-)usability in the clinic:Adequate model evaluation depends on the intended purpose of the developed model. For example, a model needs a good predictive performance for successful MIPD, while this requirement might be lower if the model is only used for investigational purposes. However, a survey of published systems biology ordinary differential equation models from 2002 to 2004 judged only 28% of models to be adequately evaluated for their intended purpose, thus posing serious difficulties in judging the suitability of many published models for further use [33].The reproducibility of developed models is dependent on the availability of model code. In a recent review, Tiwari et al. (2021) reported that only 49% of 455 published models were directly reproducible using the code and information available in the publication [34]. Even after contacting the corresponding authors, 37% of the investigated models could not be reproduced.

With this review, we aim to (i) compile and analyse published modelling activities characterising PK of IFX in UC patients, and subsequently, (ii) assess the current knowledge regarding the impact of disease activity on IFX PK in the UC population, both as a whole and in the sub-population with ASUC. As one of the first reviews on this topic, this work identifies and clarifies the knowledge gaps, thus facilitating and setting the scene for subsequent work on improving and individualising the treatment with IFX in UC and ASUC patients.

## 2. Materials and Methods

A semi-systematic literature search was conducted in April 2022, with no restrictions on publication date, to identify publications reporting population PK models of IFX for UC patients alone or comprising an IBD population.

Inclusion criteria considered for this literature research were:Report on modelling activities to characterise PK of IFX;Modelling activities, including an IBD or UC population.

Exclusion criteria were defined as follow:Modelling activities without population PK approach;Statistical approach without population PK approach (e.g., exposure response analysis without a population PK model);Populations not including UC patients.

The search was conducted in Medline (PubMed), with several word associations:Infliximab AND Ulcerative colitis AND {either Model, Modelling, Modeling or Pharmacokinetic};Infliximab AND Inflammatory bowel disease AND {either Model, Modelling, Modeling or Pharmacokinetic}.

A total of 451 articles were then screened for inclusion into the review as shown in Figure 1. From those 451 articles, 150 duplicates were removed, and the remaining 301 records were first screened through their titles and abstracts to ensure that the topic of the selected article was related to PK of IFX in populations including UC patients. The subsequent 35 selected articles were then screened by full text, and the final selection comprised 26 publications.

## 3. Results

The results of the semi-systematic research were compiled and analysed with three main focuses:**Populations included in the identified model**: The main goal of this section was to assess the place of UC and ASUC subpopulations in the modelling activities, as well as the quality of the data used for model development of IFX. Another focus of this section was the assessment of the type of data reported (e.g., disease activity, ADA)**Models of IFX**: In this section, the aim was to summarise and analyse the different type of models developed for IFX as well as the place of key factors such as ADA in the different identified original models.**Model evaluation and clinical application of developed models**: This section aimed to assess the quality of model evaluation and performance of published models. As such models are often re-used by later studies and applied in a clinical setting (e.g., MIPD), good evaluation and reporting of these evaluation methods are important.

### 3.1. Identified Gaps in the UC Populations Included in the Models

**UC subpopulations.** UC, although belonging to the family of IBD, differs in many ways from CD [2]. However, of the 26 selected articles, only four of them developed their model specifically for the UC population (15%). In addition, UC is often under-represented in the IBD populations used for modelling activities. Out of the 22 selected articles reporting on an IBD population, only eight included more than 30% of UC patients [36,37,38,39,40,41,42,43]. This does not reflect real life populations, as the prevalence of UC in the IBD population is quite similar or even higher than CD [2]. Thus, modelling of IFX PK specifically for UC patients is vastly lacking still.

Moreover, out of the 26 evaluated publications, two used a virtual population [44,45], and two used previously published data [46,47], reducing the overall number of original studies to 22 (85% of the selected articles). The original study data extracted from the selected articles are presented in Table 1.

**Disease severity.** In the 18/22 (82%) reported IBD populations used for PK modelling of IFX, disease activity of the UC sub-population remained quite similar and did not include patients with severe disease activity (median CRP concentrations, ranging between 0.4–13.3 mg/L). In addition, 4 out of 22 publications did not report the disease severity of their population [37,40,51,60].

Overall, only three studies included severe UC patients [55,56,57], as shown by higher disease scores such as fMayo score for Kevans et al. [55] (median [range], 11 [8,9,10,11,12]), SCCAI score for Berends et al. [57] (median [range], 10 [1,2,3,4,5,6,7,8,9,10,11,12,13,14,15]) or MES for Dreesen et al. [56] (48% patients with the most severe score, 3). In addition to higher disease activity scores, reported CRP concentrations for studies including severe UC patients were also higher than for non-severe UC populations (reported medians ranging between 6.1–33 mg/L for severe UC patients vs. 0.4–13.2 mg/L for non-severe UC patients), reflecting a higher inflammatory burden. Of note, out of the three original studies including severe UC patients, one comprised only 17% ASUC patients [56], and another one included moderate to severe UC patients [57]. Thus, only one study looked specifically into ASUC patients [55]. The limited number of studies including severe UC patients indicates a current knowledge gap and underlines the need for more dedicated investigation of disease severity and inflammatory burden impact on IFX PK in UC patients.

**Immunogenicity.** ADA have already been related to an increased clearance of mAbs and a higher risk of primary failure or loss of response in IBD patients receiving IFX [62,63]. However, even though ADA is almost systematically reported as shown in Table 1 (only one article did not include patients with ADA [51] and two did not report on the status [48,57]), 18 reported solely ADA status, amongst which, two measured ADA only when measured IFX concentrations were below the lower limit of quantification [38,40]. Only two selected studies reported on ADA titres [52,58], and only one reported specifically on the ADA sub-entity, neutralising ADA [41]. Thus, more thorough ways of reporting and investigating immunogenicity are needed to take into account its impact on IFX PK and LOR in UC patients.

Overall, as shown by the reported populations used for modelling activities of IFX, UC patients are often under-represented in IBD populations, and only a few studies focus on the UC sub-population. ASUC patients, with a high inflammatory burden, are underrepresented. In addition, disease activity is not often taken into account. Thus, consequences of disease activity and severity on IFX PK are currently poorly characterised.

### 3.2. Current Available Models for IFX in UC/IBD Patients

In the conducted semi-systematic literature search, 14 original population PK models describing IFX PK disposition were identified for populations that included UC patients (Table 2). UC patients are most commonly treated with intravenous infusion administration of IFX, which was reflected in the identified studies. All studies described intravenous administration except for the study by Hanzel et al. [41], in which IFX PK after subcutaneous administration was described, and the bioavailability was estimated to be 79.1% with an absorption rate of 0.273 day^−1^.

The majority of the 14 identified models were standard one- or two-compartment models with first-order elimination. However, Berends et al. [57] expanded on a standard two-compartment model by incorporating a target-mediated drug disposition (TMDD) model describing the interaction of IFX and TNFα under a quasi-steady state approximation. The inclusion of TMDD allowed for the recapitulation of the increasing concentrations of TNFα often observed in the start of IFX therapy by prolonging the TNFα half-life through complex formation with IFX.

Overall, PK parameters estimated by the 14 models remained within the expected values for IFX. Volumes of distributions, either for one compartment model (solely central volume of distribution) or two compartment models (as the sum of both peripheral and central volumes of distribution) ranged from 2.1 L for the Ternant model [48] (but solely for women) to 11.5 L for the Petitcollin model [38]. Regarding clearance, except for a high clearance found by Ternant et al. of 0.768 L/day, for ADA-positive patients, the estimated values ranged from 0.199 L/day [50] to 0.407 L/day [49]. Of note, the highest estimated clearances were found for models developed solely for UC patients [49,57], eluding to the idea that disease may have a significant impact on clearance, and IFX PK for UC or CD patients may not show the same characteristics. Clearances estimated solely for UC patients ranged from 0.330 to 0.407 L/day, whereas clearance estimated for an IBD population ranged from 0.199 to 0.358 L/day. The most common identified covariate on volume of distribution was weight (57% of published models). The remaining unexplained interindividual variability for the volumes of distribution varied amongst the different models, ranging from 6.86% to 76.1%. For clearance, more covariates were often identified as impactful, amongst which the most commonly included were ADA (10/14 of models) and serum albumin concentrations (9/14 of models). The interindividual variability associated with clearance was overall lower than the one for volumes of distribution with values ranging from 13.6% to 44.3%. All those parameters are reported in Table 2, which included the reported residual unexplained variability for consistency. In addition, the impact of identified PK parameters in the original models is summarised in Figure 2.

Elimination of IFX was included as a first-order process except for two published models. Both Kevans et al. [55] and Petitcollin et al. [38] developed PD models including time-dependent clearance: Kevans et al. identified an inverse association between time on IFX therapy and IFX clearance, with the highest clearance during the induction phase of therapy. However, the presence of ADA inverted the time dependency, resulting in increasing IFX clearance over time. As increased IFX clearance can be a driving factor of LOR, considering such changes over time is of great importance to improve IFX therapy in UC patients. Petitcollin et al. also described increased IFX clearance with time regardless of ADA status while including a logit-risk model of immunization (Appendix A). Only one patient showed an ADA-positive test; however, 11/93 patients showed a rapid change in IFX clearance. The inclusion of the logit-risk model significantly improved the predictions and could capture the observed rapid changes in IFX clearance.

The presence of ADA has been identified as an important factor underlying increasing IFX clearance. Nine out of the fourteen identified models reported ADA status as a significant covariate on clearance, with an associated increase ranging from 21–167%. None of the models included observed ADA titres as a continuous covariate. However, Brandse et al. [52] linked a two-compartment IFX model with an ADA time-to-event model, describing the probability of developing ADAs as driven by a time-overestimated IFX trigger concentration of 3 µg/mL [52]. The time-to-event model was linked to an ADA PD model describing ADA dynamics after onset of production. The estimated titres were subsequently linked with the IFX PK model by increasing IFX clearance. This model constitutes a step towards understanding the relationship between ADA dynamics and IFX PK.

Four out of the fourteen identified original models were repurposed and used in other analyses. The re-estimated parameters are reported in detail in Appendix A. To facilitate the reuse of models, the availability of model code and/or clearly specified model structure is imperative. Only two of the original models provided the full model code as Appendix A [56,57]. The associated ordinary differential equation system and/or model schematic was provided for four models. The specific implementation of covariates and random effects was reported for 12 and 9 models, respectively.

### 3.3. Model Evaluation and Current Clinical Application of Developed Models

In order to apply a model in the clinical setting, the goodness of fit (GOF) and predictive performance should be assessed [31,66]. In Table 3, an overview of the model evaluation techniques applied in the 26 identified studies is provided. Overall, model evaluation is often reported very concisely, and the applied evaluation techniques are often basic; in 5/26 (19%) studies, no model evaluation is reported at all. These papers refer to previous work even if the model is applied to a new population. Almost half (11/26) of the studies do not report basic model evaluation such as GOF plots. Of note, two of these studies mention GOF plots in their respective method section but do not show any results. Similarly, 10/26 studies do not report advanced model evaluation results, making judgment of the predictive performance of the applied model impossible. This is especially problematic when the model is used in a predictive setting, e.g., for dose individualization. Of the 16 studies reporting advanced model evaluation methods, nine apply only one single method, while the latest recommendations for the reporting of population pharmacokinetic analyses [67] require the use of multiple methods to assess both model robustness and predictive performance. The visual predictive check (VPC) is by far the most-reported method for assessment of predictive performance. Often cited as the golden standard in population PK model evaluation [68], the VPC compares prediction intervals based on stochastic simulations to the observed variability in the original dataset.

Half of the studies (13/26) described the application of a previously developed model in a new setting. Almost half of these (6/13) applied this model to a new population for, e.g., Bayesian forecasting, without evaluating the predictive performance of the model.

Model performance was reported as adequate by the authors in most studies that performed model evaluation, with bias being typically present for the lower concentrations, although this was often masked by the use of GOF plots on a normal scale and not on a logarithmic scale. For ADA-positive patients, model evaluation performed by Schräpel et al. on previously developed models showed a poor predictive performance for 7 of the 26 selected models. This raises the questions about predictive performance of ADA-positive patients for the remaining models, which was not evaluated.

The links between the 26 selected articles from the literature research (as shown in Figure 1), the 14 original models developed in those articles, as well as the different selected articles applying those original models are summarised in Figure 3.

## 4. Discussion and Perspectives

The aim of this review was to identify gaps in current knowledge and needs for improvement in pharmacometric activities for IFX PK in UC and ASUC patients as well as the applicability of those model to a clinical setting. Firstly, UC population and especially ASUC patients are highly overlooked even though these populations are in need of treatment optimization [69] and are more common than CD patients [2]. Moreover, most of the population PK models, published between 2008 and 2022, used to describe IFX PK are of similar structure, with a simple one- or two-compartment model and first order elimination. This contrasts with what can be found for most mAbs, where currently more mechanistic models are used to describe their PK, including non-linear elimination. When looking at general PK of mAbs, TMDD models and their approximations seem to be more appropriate to describe the biological mechanism process of therapeutic antibodies [70]. However, in order to be able to develop such models, measurements of their target are needed, which are lacking in most data reported in the selected articles. Among the identified original studies, only two original datasets reported on IFX target concentrations, TNFα [50,57]. Out of the two models reporting TNFα concentrations, only one used it for modelling purposes [57]. TNFα is a well-known cytokine, and is measured in other diseases, especially as a biomarker of inflammation [71,72,73]. Thus, its incorporation in modelling activities for IFX could be a way forward.

Immunogenicity has been shown to impact mAbs PK. The formation of immune complexes was shown to increase mAb clearance, dependent on ADA levels [62,63]. However, ADA are mostly reported and included in the models as a positive or negative status. The use of ADA levels as a continuous covariate could greatly improve the prediction of IFX PK, especially clearance, for ADA positive patients. Edlund et al. showed, for CD patients, that including ADA as a continuous covariate improved PK predictions, as ADA concentrations appeared to be related to an increase in IFX clearance [74]. In the selected articles, only two models took ADA into account in a more mechanistic way, seeking to implement immunogenicity impact on clearance and improving model predictions [38,52]. In general, as shown by Schräpel et al., poor predictions are observed for ADA-positive patients compared to ADA-negative in most developed models [61]. Thus, implementation of more mechanistic models, accounting for the impact of ADA development on PK, as well as on pharmacodynamics, could improve the therapeutic management of ADA-positive patients. In this regard, more complex modelling approaches have been investigated to describe immunogenicity of mAbs and their intertwined relationship with PK and PD [75,76,77].

Another issue identified was the lack of disease activity factors included in modelling activities. Even though disease scores are often reported, the inflammatory burden of those patients has not been included in the models. It has been shown that a severe disease state, leading to a high inflammatory burden in the gut, could increase IFX clearance. This could lead to a possible non-response of patients, as early as the induction phase of the treatment [19]. Three models took CRP concentrations into account, either as a covariate impacting clearance [38,56] or as a biomarker for PD [59], while three models directly investigated disease scores’ impact on PK [38,50,56]. Thus, 4 out of the 14 original population PK models took disease activity into account. Petitcollin et al. included both CRP concentrations and disease score as impactful covariates on clearance. However, even though an IBD population was investigated, only the pMayo score, specific for the UC subpopulation, was found as a significant disease activity covariate on clearance. This eludes to the fact that UC patients might have a greater inflammatory burden, leading to a greater impact of disease severity on IFX PK. This could also be explained by a better relation of disease activity indices to inflammatory burden in UC than in CD, where fibrostenotic disease might have a high impact on symptom-based disease scores while the associated inflammation is limited [78,79]. Thus, specifically considering inflammation, especially in ASUC patients, could help clinicians to better adapt an IFX dosing regimen in severe UC.

Furthermore, some limitations in methodology, model performance and applicability of the published models have been identified. Out of the 26 selected articles, only 14 models were newly developed. The parameters from Fasanmade’s models, either for UC patients [49] or CD patients [64], were often re-estimated based on a new dataset (four and seven times, respectively). The same applies for Xu et al., which was re-used five times. Noteworthy, only an abstract with limited information about the model is available [65], thus lacking relevant and thorough model specification and evaluations. The models by Fasanmade et al. were developed on data from a high number of patients (482 UC patients and 682 CD patients for Fasanmade 2009 and Fasanmade 2011, respectively) emerging from clinical trials, with rich sampling of IFX concentrations. However, models using data from clinical routine, often reporting only C_min_, used these two models to predict IFX concentrations in their own populations and updated the parameters of the Fasanmade model accordingly without taking the difference in information content between a densely sampled large clinical study dataset and a sparsely sampled small observational TDM dataset into account. Furthermore, as the Fasanmade models were developed using specific populations (either UC or CD), the question of the applicability of one model to an IBD population comprising both UC and CD patients can be raised. As shown by Passot et al., disease type can have an impact on IFX clearance, and thus, use of models developed for a different population should be performed with care. Lastly, as mentioned, most original data from the selected publications emerged from a clinical setting, often reporting solely sparse C_min_ from a retrospective study. However, as shown by the different model evaluation reported in Table 3, predictions of low concentrations are often biased. As most clinical settings used C_min_ as a target for treatment efficacy [80], this raises the question of the appropriateness of dosing regimen recommendations based solely on predicted C_min_ by those models.

What also emerged from the reported model evaluation is that predictive performance of published models is either poorly reported or insufficient. In addition, lack of information supplied in manuscripts as well as the absence of an available model code make reproducibility of published models challenging. As a consequence, despite many reported models for IFX in IBD or UC patients, clinical applicability of most selected models, e.g., for MIPD, is not possible. The fact that this aspect emerged as crucial in recent years might explain this gap for the first published models. More recent publications such as the ones from Konecki et al. [60] and Schräpel et al. [61] show a growing effort to adequately evaluate model performance. However, this remains important, especially regarding ASUC patients who are in need of dose optimization from the start of treatment to ensure a fast clinical improvement and prevent colectomy. In general, it is the responsibility of the modelling community to follow a good practice in reporting developed models to enrich scientific knowledge in the field and, in the end, improve patient care in a clinical setting. However, editorial and reviewing bodies can also aid by requiring more exhaustive reporting of the model (i.e., model code) and the evaluation of predictive performance, which can easily be provided in online Appendix A.

These findings show a lack of appropriate models for MIPD for UC patients receiving infliximab therapy. Models developed in the early 2010s have been used many times for the past 10 years with few improvements or further investigation on factor-influencing IFX PK. ADA have been mainly included as a categorical covariate even though the mechanistic understanding of their development, as well as their interaction with IFX, would greatly improve management of ADA-positive patients, which in some studies can represent up to 60% of the population treated by IFX [81]. In addition, model evaluations of the current published models are not extensive or are simply not shown, putting into question their true predictability. Overall, reporting of model code and extensive evaluation are lacking. Reproducibility of many published models is thus impossible, limiting the possibility to apply these models for clinical dose optimisation. Given the amount of models and data published for IFX, the use of model-based meta-analyses could be useful in a dosing individualization setting. This methodology aggregates data and information extracted from systematic literature research [82,83]. Leveraging the knowledge from all published models could lead to an appropriate description of the drug’s PK and PD and thus considerably help with dosing optimisation in this population. However, the use of these techniques is dependent on both transparently demonstrated adequacy of model performances and reproducibility of the published work.

## 5. Conclusions

In conclusion, there is a need for appropriate model development of IFX for UC patients, especially ASUC patients. UC patients are often underrepresented in IBD populations, and only few studies focus on this sub-population. Developed models should take into account the inflammatory load, for example assessed by CRP concentrations, endoscopic assessment, and disease scores to ensure that IFX clearance is properly estimated for those patients. A comprehensive model evaluation should be conducted in order to then apply this model in a clinical setting to optimise the dosing regimen of IFX in severe UC patients as early as possible in the therapy. In the end, adequate development and reporting of models taking into account key factors in IFX PK for UC and ASUC patients could improve clinical outcome of those patients, leading to early remission and preventing colectomy.

## Figures and Tables

**Figure 1 pharmaceutics-14-02095-f001:**
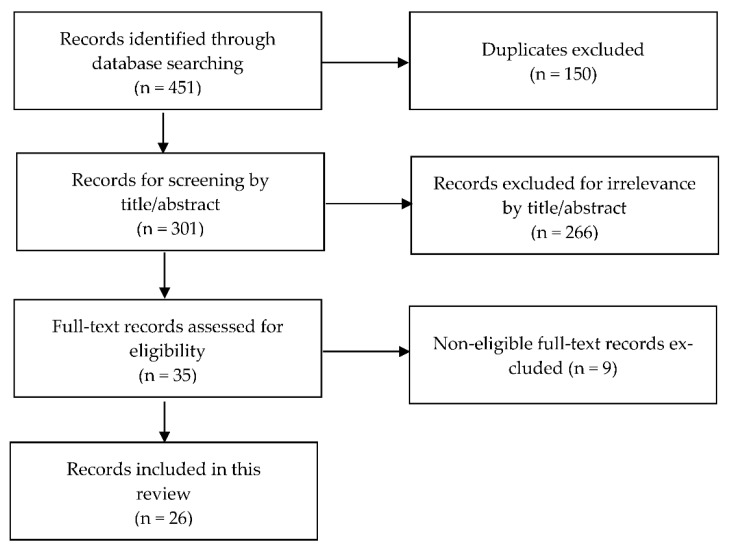
Selection process of records reporting on population pharmacokinetic models for infliximab in ulcerative colitis patients. The flowchart was based on the Preferred Reporting Items for Systematic Reviews and Meta-Analyses (PRISMA) guidelines [35].

**Figure 2 pharmaceutics-14-02095-f002:**
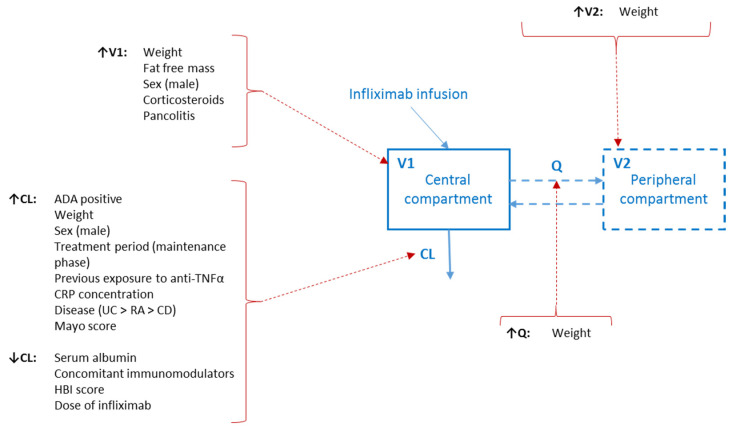
Impact of identified covariates in the 14 models on infliximab PK parameters. As one- or two-compartment PK models were developed, the second compartment and the associated mass transfer process are displayed with dashed lines. **Abbreviations:** ADA: Anti-drug antibodies; CD: Crohn’s disease; CL: Clearance; CRP: C-reactive protein; HBI: Harvey–Bradshow index; Q: Intercompartmental clearance; RA: Rheumatoid arthritis; TNF: Tumour necrosis factor; UC: Ulcerative colitis; V1: Volume of distribution of the central compartment; V2: Volume of distribution of the peripheral compartment.

**Figure 3 pharmaceutics-14-02095-f003:**
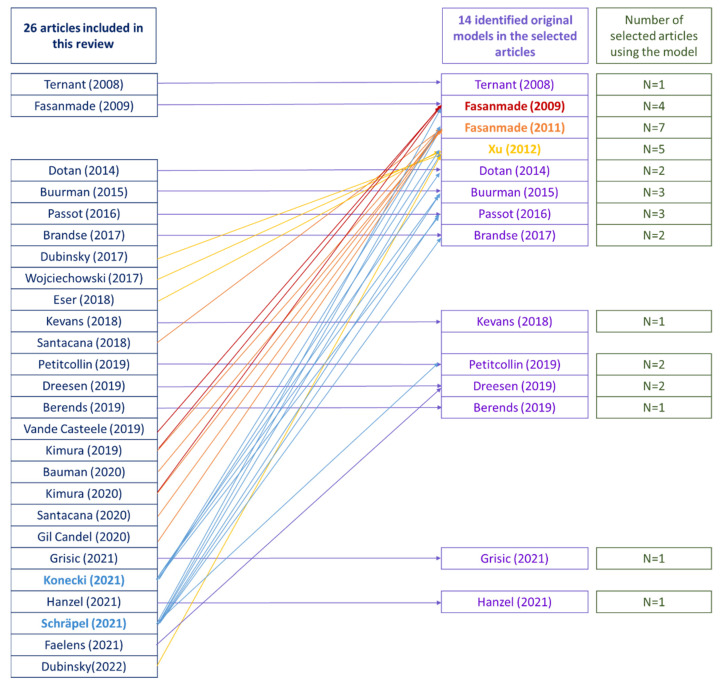
Scheme of the 26 selected articles linked to one or several of the identified original models. Arrows link the selected articles to the original model used in their study. Konecki (2021) [60] and Schräpel (2021) [61], in blue, are displayed in a different colour, as their studies consisted in evaluated published models, thus including several original models in their publication. Fasamnade (2009) [49], in red, Fasanmade (2011) [64], in orange, and Xu (2012) [65] in yellow are displayed in different colours, as their original models were extensively re-used in studies from the selected articles.

**Table 1 pharmaceutics-14-02095-t001:** Patient populations included in the selected articles and clinically relevant characteristics.

Original Study Population [Reference]	Total Number of Patients: Subpopulations: N (%)	Study Type IFX Sample Type	Clinical Evaluation (e.g., Reported Score)	Endoscopy Assessment (Yes/No)	Montreal Classification Assessment	CRP Concentrations, Median (Range)	Serum Albumin Concentrations, Median (Range)	Weight, Median (Range)	ADA Assessment	Other Relevant Disease Factors Reported
**Ternant (2008)** [48]	33 patients: UC: 3 (9%) CD: 30 (91%)	Retrospective study, C_min_ and 2 h post end of infusion	CDAI and CRP	No	No	Determined, NR	No	67 kg (44–110 kg)	Status: 5/33 positives (15%)	Concomitant IMM
**Fasanmade (2009)** [49]	482 patients: UC: 482 (100%)	Prospective clinical trial, Rich PK sampling	fMayo score	Yes, in fMayo score	No	8.0 mg/L (2.0–227)	41 g/L (24–52)	77 kg (40–177.3)	Status: 33/482 positives (6.8%)	Concomitant IMM Liver and kidney functions Blood cells
**Dotan (2014)** [36]	54 patients: UC: 25 (46%) CD: 25 (46%) Undetermined IBD: 4 (8%)	Prospective study, C_min_	CD: CDAI (135 ± 86.5) UC: pMayo (n = 24, 2.7 ± 1.4) or fMayo (n = 11, 5.7 ± 3.9) PGA: 4 patients	Yes, in fMayo score (n = 11)	Partial (location only)	Mean ± SD 15.2 mg/L ± 16.9 mg/L	Mean ± SD 40 g/L ± 5 g/L	Mean ± SD 68.5 kg ±14.8 kg	Status: 17/54 positives (32%)	Disease duration Concomitant medications Smoking status Sex
**Buurman (2015)** [50]	42 patients: UC: 8 (19%) CD: 34 (81%)	Retrospective study, C_min_	HBI: 6 (3–24) GPA: 1 (1–3)	No	Determined, NR	5 mg/L (5–105)	41 g/L (33–50)	74 kg (51–145)	Measured at week 54 only, as status: 2/42 positives (5%)	Concomitant IMM TNFα Leucocytes
**Passot (2016)** [51]	218 patients: UC: 16 (7%) CD: 63 (29%) AS: 91 (42%) RA: 18 (8%) PsA: 30 (14%)	Retrospective study, C_min_	No	No	No	No	No	67 kg (28.2–125)	Patients with ADA not included	Methotrexate
**Brandse (2017)** [52]	332 patients: UC: 79 (24%) CD: 253 (76%)	Retrospective study, Sparse reactive TDM sampling	Disease extent of Montreal classification	No	Disease extent only (UC patients: 8% proctitis, 38% left colitis, 46% pancolitis)	No	Mean ± SD 35.2 g/L ± 15.2 g/L	Mean ± SD 72.3 kg ± 16.3 kg	Titres: 2 (1–16)	Anti-TNFα naïve (80%) Concomitant medications Disease duration
**Dubinsky (2017)** [53]	50 paediatric (7 to 20 years) patients: UC: 9 (18%) CD: 41 (82%)	Retrospective study, C_min_	Paediatric CDAI or CDAI pMayo	No	No	2.07 mg/L (0.020–12.0)	39.1 g/L (26–48)	40 kg (19–98)	Status: 14/50 positives (28%)	Concomitant medications Complete blood count
**Eser (2018)** [54]	117 patients: CD: 86 (73%) UC: 30 (26%) Unknown: 1 (1%)	Prospective study, C_min_	pMayo: 2 (0–11) HBI: 1 (0–6)	No	No	2.8 mg/L (0.3–74.9)	43.2 g/L (25.3–50.8)	70 kg (47–130)	Status: 19/117 positives (16%)	Concomitant medications Smoking status Sex
**Kevans (2018)** [55]	ASUC cohort: 36 patients (100%) and Validation PK cohort: 51 patients UC: 42 (82%) CD: 9 (18%)	ASUC cohort Retrospective study, Rich sampling during induction phase	fMayo: 11 (8–12)	Yes, in fMayo score	No	33 mg/L (1–240)	33 g/L (24–48)	61 kg (41–104)	Status: 6/36 positives (17%)	Hospitalisations Disease duration
**Santacana (2018)** [37]	100 patients: UC: 34 (34%) CD: 66 (66%)	Prospective study, C_min_	No	No	No	1.8 mg/L (0–170.4) CD: 1.9 mg/L (0–170.4) UC: 1.55 mg/L (0–20.6)	43 g/L (23–53) CD: 43 g/L (23–53) UC: 43 g/L (37–53)	72 kg (41–122)	Status: 21 positives samples (of 370, 5.7%)	Concomitant IMM
**Petitcollin (2019)** [38]	91 patients: UC: 29 (32%) CD: 62 (68%)	Prospective study, C_min_	UC: pMayo CD: HBI	Yes, but not for all	Yes	NS	No	66 kg (60–69)	Measured if C_min_ < 0.1 µg/mL Only one positive sample	Previous treatment Disease duration Smoking status Sex Blood cell count
**Dreesen (2019)** [56]	204 UC patients ASUC: 34 (17%)	Retrospective study, C_min_	MES 0: 0 (0%) 1: 7 (3.4%) 2: 96 (47.1%) 3: 98 (48%) NA: 3 (1.5%)	Yes, as mucosal healing: Yes: 91 (45%) No: 74 (36%) NA: 39 (19%)	Disease extension: E3 (pancolitis) 123 (60%)	6.1 mg/L (IQR: 2.4–19.9)	42 g/L (IQR: 38.8–43.9)	72 kg (IQR: 61–82)	Status only for samples with undetectable IFX: 7/605 (1%)	Disease duration
**Berends (2019)** [57]	20 patients with moderate to severe UC	prospective study, Rich sampling	fMayo (95% score = 3) SCCAI: 10 (1 –15)	Yes, in fMAYO	Extent: Left side colitis (35%) pancolitis (65%)	25.3 mg/L (0.6–196.2)	38 g/L (23–45)	70 kg 47–90)	Status: 7/20 positives (35%)	Sex Disease duration Hospitalisation Concomitant thiopurines FCal TNFα
**Bauman (2020)** [58]	228 paediatric patients Model development cohort (14.5 ± 3.6 years): n = 135 UC: 26 (19%) CD: 109 (81%) Validation cohort (13.3 ± 3.8 years): n = 93 17 UC: 17 (18%) CD: 63 (68%) Unclassified: 13 (14%)	Retrospective study, C_min_	Discovery cohort only UC: PUCAI 20 (6.3 –33.8) CD: sPCDAI 20 (10 –35) PGA for all: Quiescent: 17 (57%) Mild: 10 (33%) Moderate: 3 (10%) Severe: 0 (0%)	No	No	Discovery: 2.9 mg/L (2.9 –4.6) Validation: 2.9 mg/L (2.9–6-3)	Mean ± SD Discovery: 37.7 g/L ± 5.1 g/L Validation: 37.7 g/L ± 4.2 g/L	Mean ± SD Discovery: 56 ± 22 kg Validation: 51 ± 20 kg	Status: Discovery: 62% + Validation: 50% + Concentrations as 4 ordinal categories (from 22 to 1000 ng/mL)	Sex Blood cell count Haematocrit
**Kimura (2020)** [43]	15 patients: UC: 7 (47%) CD: 8 (53%)	Prospective study, C_min_ and in between doses (prospective TDM)	UC: pMayo 7.14 ± 1.46 (5–9) CD: CDAI 244 ± 45 (166.5–326.1)	No	No	No	Mean ± SD (range) 36.9 g/L ± 7.1 g/L (21–47)	Mean ± SD (range) 58.6 kg ± 22.7 kg (34–76)	According to Figure 1 only, status: 6/15 positives (40%) 14/85 samples positives (17%)	Concomitant IMM
**Santacana (2020)** [39]	108 patients: UC: 36 (33%) CD: 72 (67%)	Retrospective + prospective study, C_min_	UC: pMayo 1 (0–4) CD: HBI 1 (0–5) Biological response (CRP ≤ 5 mg/L)	No	No	All: 1.8 mg/L (1–4.45) UC: 1.05 mg/L (0.95–3.5) CD: 1.95 mg/L (1.0–5.4)	All: 43.5 g/L (42 –46) UC: 44 g/L (41.8–46) CD: 43 g/L (42–46)	71 kg (60–82)	Status: 17/108 positives (16%) Patients that discontinued treatment: 15/16 positives (93.8%)	Concomitant IMM Sex FCal
**Gil Candel (2020)** [40]	TDM: 47 patients UC: 12 (32%) CD: 35 (68%) Evaluation: 50 patients UC: 12 (24%) CD: 38 (76%)	Retrospective study, C_min_	NR: based on previously established IFX threshold: 3 µg/mL for CD and 5 µg/mL for UC	No	No	TDM-A (n = 21): 0.4 mg/L (IQR: 0.2) TDM-B (n = 26): 0.4 mg/L (IQR: 0.1) Evaluation: NR	TDM-A (n = 21): 43 g/L (IQR: 4) TDM-B (n = 26): 44 g/L (IQR: 5) Evaluation: All: 43 g/L (IQR: 4) UC: 45 g/L (IQR: 5) CD: 43 g/L (IQR: 5)	TDM-A (n = 21): 70 kg (IQR: 18) TDM-B (n = 26): 73.5 kg (IQR: 23) Evaluation: All: 70.9 kg (IQR: 18.5)	Measured if Cmin < 1 µg/mL Evaluation: 2/50 positives (4%), 1 UC and 1CD	Sex FCal Concomitant IMM
**Grisic (2021)** [59]	121 patients: UC: 31 (26%) CD: 89 (74%) Undetermined: 1 (<1%)	Prospective study, C_min_	HBI: 2 (0–19)	No	Yes Severity UC: Mild: 6% Moderate: 28% Severe: 21%	2.70 mg/L (0.2–120)	42.9 g/L (25.3–51.6)	70 kg (47–115)	Status: 82/388 positives samples (21%)	Sex Smoking status Disease duration
**Konecki (2021)** [60]	157 patients: UC: 18 (12%) CD: 116 (74%) AS: 22 (14%) RA: 3 (2%) PsA: 3 (2%)	Retrospective study, Sparse TDM samples	“Disease scores” NS	Determined, NR	No	Determined, NR	Determined, NR	68 kg (24–150)	Status: 12/157 positives (7.6%)	Concomitant IMM Sex Age Smoking status
**Hanzel (2021)** [41]	175 patients: UC: 78 (45%) CD: 97 (55%)	Prospective clinical trial, Rich PK sampling	fMayo: 8 (4–11) CDAI: 283 (249–433)	Yes, in fMayo score	No	3.2 mg/L (0.2–89.4)	44 g/L (28–54)	69 kg (43–118)	Nab status: 58/175 positives (33%)	Sex Age Concomitant IMM Corticosteroids WBC Platelets FCal
**Schräpel (2021)** [61]	105 patients: UC: 29 (28%) CD: 76 (72%)	Prospective study, C_min_ and midpoint samples	HBI: 1 (0–18)	No	No	2.9 mg/L (0.20–74.9)	43:5 g/L (25.3–50.8)	70 kg (47–115)	Status: 22/105 positives (21%) 49/336 positives samples (15%)	Concomitant IMM Smoking status Sex
**Dubinsky (2022)** [42]	180 patients: UC: 55 (31%) CD: 129 (69%) Stratified by previous IFX dose: INF1: 5 mg/kg (n = 95) INF2: 10 mg/kg (n = 85)	Prospective study, C_min_	pMayo: 6 (IQR: 4–8) HBI: 2 (IQR: 0–7)	No	Yes, only location shown	INF1: 7.5 mg/L (3.1–24.5) INF2: 13.3 mg/L (2.9–41.7)	INF1: 3.4 g/L (31–39) INF2: 29 g/L (25–35)	INF1: 44.5 kg (30.5–66.1) INF2: 49.8 kg (39.5–63.1)	Status: 23/180 positives (13%)	Sex Age Disease duration Concomitant IMM and/or steroids

ADA: Antidrug antibodies; AS: Ankylosing spondylitis; CD: Crohn’s disease; Cmin: Minimum concentrations; CRP; C-Reactive protein; FCal: Faecal calprotectin; fMayo: Full Mayo score; GPA: Global physician assessment score; HBI: Harvey–Bradshaw index; IBD: Inflammatory bowel diseases; IFX: Infliximab; IMM: Immunomodulators; INF: Infusion; IQR: Interquartile range; MES: Mayo endoscopic score; Nab: Neutralising ADA; NR: Not reported; PD: Pharmacodynamics; PGA: Physician global assessment; PK: Pharmacokinetic; pMayo: partial Mayo score; PsA: Psoriatic arthritis; PUCAI: Paediatric ulcerative colitis activity index; RA: Rheumatoid arthritis; SCCAI: Simple Clinical Colitis Activity Index; SD: Standard deviation; TNF: Tumour necrosis factor; sPCDAI: Short paediatric Crohn’s disease activity index; UC: Ulcerative Colitis; WBC: White blood cells.

**Table 2 pharmaceutics-14-02095-t002:** Selected population pharmacokinetic models for infliximab in ulcerative colitis or inflammatory bowel diseases patients.

Structural PK(-PD) Model						Covariate Model	IIV/IOV Model	Residual Model	Specified Implementation	
*Parameter estimates (RSE)*						*Parameter estimate (RSE)*	*Parameter estimate as CV (RSE)*	*Error type:* *parameter estimate (RSE)*		
**Ternant (2008)** [48]: 2-CMT model, first order elimination										
*CL (L/h)*		*V1 (L)*	*V2 (L)*		*Q (L/h)*	NA, see structural model	CL ADA_−_: 22.5%	Add.: 1.04 mg/L (12.0%)	*Full model code*	NO
ADA_−_: 0.012 (7.9%)		SEX_M_: 2.3 (18.1%)	1.0 (22.6%)		0.0054 (34.5)		CL ADA_+_: 22.7%		*ODEs/model schematic*	NO
ADA_+_: 0.032 (10.9%)		SEX_W_: 1.1 (31.4%)					V1 SEX_M_: 14.1%		*Covariate relationships*	YES
		WT: 1.7 (19.6%)					V1 SEX_W_: 11.3%		*Random effects*	YES
							V1 WT: 1.719.3%			
							V2: 15%			
							Q: 10%			
**Fasanmade (2009)** [49]: 2-CMT model, first order elimination										
*CL (L/d)*		*V1 (L)*	*V2 (L)*		*Q (L/d)*	ALB~CL: −1.54 (2.5%)	CL: 37.68% (8.5%)	Prop.: 40.30% (2.6%)	*Full model code*	NO
0.407 (2.5%)		3.29 (2.1%)	4.13 L (3.9%)		7.14 (6.8%)	ADA_cat_~CL: 0.471 (22.5%)	V1: 22.11 (16.6%)	Add.: 0.0413 µg/L (3.5%)	*ODEs/model schematic*	NO
						SEX_W_~CL: −0.236 (11.9%)			*Covariate relationships*	YES
						WT~V1: 0.538 (13.5%)			*Random effects*	YES
						SEX_W_~V1: −0.137 (23.2%)				
**Fasanmade (2011)** [64]: 2-CMT model, first order elimination										
*CL (mL/kg/d)*		*V1 (mL/kg)*	*V2 (mL/kg)*		*Q (mL/kg/d)*	ALB~CL: –0.855 (13.5%)	CL: 30.7% (8.1%)	Prop.: 29.2% (2.7)	*Full model code*	NO
5.42 (2.0)		52.4 (0.9)	19.6 (4.2)		2.26 (9.9)	ADA_cat_~CL: 0.291 (15.2%)	V1: 12.6% (19.2%)	Add.: 0.371 µg/mL (18.3%)	*ODEs/model schematic*	NO
						IMM~CL: –0.137 (20.5%)	V2: 55.3% (18.5%)		*Covariate relationships*	YES
						WT~CL: –0.313 (14.6%)			*Random effects*	YES
						WT~V1: –0.233 (12.3%)				
						WT~V2: –0.588 (16.1%)				
**Xu (2012)** [65]: 2-CMT model, first order elimination										
*CL (L/h)*		*V1 (L)*	*V2 (L)*		*Q (mL/kg/d)*	WT~CL: 0.612 (4.7%)	CL: 31.3% (6.9%)	Prop.: 41.9% (1.0%)	*Full model code*	NO
0.296 (1.5%)		3.3 (1.1%)	1.16 (4.9%)		0.0781 (11.4%)	ALB~CL: −2.3 (9.3%)	V1: 9.85% (33.9%)		*ODEs/model schematic*	NO
						ADA_cat_~CL: 0.231 (24.7%)	V2: 76.1% (12.3%)		*Covariate relationships*	YES
						WT~V1: 0.696 (2.9%)	Q: 111% (14.4%)		*Random effects*	YES
						WT~V2: 0.604 (19.5%)				
						WT~Q: 1.15 (21.2%)				
**Dotan (2014)** [36]: 2-CMT model, first order elimination										
*CL (L/d)*		*V1 (L)*	*V2 (L)*		*Q (L/d)*	WT~CL: 0.612 (2.1%)	CL: 13.45 (4%)	NR	*Full model code*	NO
0.381 (0.5%)		2.37 (0.2%)	1.37 (0.4%)		0.122 (0.4%)	ALB~CL: −1.39 (1.5%)	V1: 42.07 (1.5%)		*ODEs/model schematic*	NO
						ADA_cat_~CL: 1.59 (13%)	V2: 32.40 (1.7%)		*Covariate relationships*	YES
						WT~V1: 0.696 (1.9%)	Q: 85.15 (1.2%)		*Random effects*	YES
						WT~Q: 1.15 (3.6%)				
						WT~V2: 0.604 (10.2%)				
**Buurman (2015)** [50]: 2-CMT model, first order elimination										
*CL (L/d)*		*V1 (L)*	*V2 (L)*		*Q (L/d)*	Period~CL: +40% (11%)	CL: 18% (18%)	Prop.: 21.7% (30%)	*Full model code*	NO
0.199 (6%)		4.94 (10%)	3.13 (32%)		0.0618 (23%)	ADA_cat_~CL: +72% (35%)	V1: 17.1% (31%)	Add.: 0.98 mg/L (18%)	*ODEs/model schematic*	NO
						SEX_M_~CL: +35% (34%)			*Covariate relationships*	YES
						HBI~V1: −3.6% (28%)			*Random effects*	NO
**Passot (2016)** [51]: 1-CMT model, first order elimination										
*CL (L/d)*		*V1 (L)*				SEX_M_~V: 0.209 (45%)	V: 0.224 (12%)	Prop.: 0.223 (7%)	*Full model code*	NO
0.23 (4%)		5.2 (4%)				≤15 y.o.~V: −0–396 (37%	CL: 0.304 (7%)	Add.: 0.72 mg/L (19%)	*ODEs/model schematic*	NO
						CD~V: 0.399 (17%)			*Covariate relationships*	YES
						Wt~CL: 0.603 (18%)			*Random effects*	YES
						SEX_M_~CL: 0.181 (30%)				
						CD~CL: 0.384 (15%)				
						UC~CL: 0.472 (21%)				
						RA~CL: 0.392 (32%)				
						MTX~CL: −0.336 (52%)				
**Brandse (2017)** [52]: 2-CMT model, first order elimination										
*CL (L/d)*		*V1 (L)*	*V2 (L)*		*Q (L/d)*	WT~CL: 0.523	CL: 38.1%	NR	*Full model code*	NO
0.358		4.72	2.4		0.0697	ALB~CL: −1.38	V1: 68.6%		*ODEs/model schematic*	NO
						Previous exposure~CL: 0.0521	V2: 71.7%		*Covariate relationships*	NO
						ADA_cat_~CL: 0.601	Q: 58.1%		*Random effects*	NO
						WT~Vc: 0.473			*Full model code*	
						WT~Q: 0.523				
						WT~Vp: 0.473				
**Kevans (2018)** [55]: 2-CMT model, time-varying CL										
*CL (L/d)*		*V1 (L)*	*V2 (L)*		*Q (L/d)*	ADA_cat_~CL_time_: 0.138 (25.5%)	CL: 50.10% (1.2%)	Prop. error: 28.5% (0.3%)	*Full model code*	NO
0.368 (0.5%)		3.3 (0.2%)	3.42 (0.5%)		0.308 (0.1%)	WT~CL: 0.709 (0%)	V1: 35.92% (0.8%)		*ODEs/model schematic*	NO
CL_time_: 0.105 (0.4%)						ADA_cat_~CL: −0.0373 (4.9%)	V2: 75.70% (0.8%)		*Covariate relationships*	NO
						ALB~CL: −0.445 (2.6%)	Q: 55.14% (0.4%)			
						WT~V1: 0.64 (0%)			*Random effects*	NO
						WT~V2: 0.991 (0%)				
						WT~Q: 1.52 (0%)				
**Petitcollin (2019)** [38]: 1-CMT model, time-varying CL										
*CL_base_ (L/d)*		*Slope (L/d/year)*		*V1 (L)*		UC~CL: 0.377 (29%)	V: 25.4% (15%)	Prop.: 20.6% (6%)	*Full model code*	NO
0.273 (7%)		0.0348 (9%)		11.5 (5%)		Dose~CL: −0.267 (22%)	CL_base_: 44.3% (10%)	Add.: 0.446 ug/mL (22%)	*ODEs/model schematic*	NO
						CRP~CL: 0.0654 (15%)	Dose~CL: 33.3% (19%)		*Covariate relationships*	YES
						Mayo score~CL: 0.0934 (30%)	slope: 32% (26%)		*Random effects*	YES
						AZA~CL: 0.849 (2%)				
						WT~Slope: 31.1 (31%)				
**Berends (2019)** [57]: 2-CMT model, first order elimination										
*CL (L/d)*		*V1 (L)*	*V2 (L)*		*Q (L/d)*	ADA_cat_~CL: 2.15 (12%)	CL: 29.2% (19%)	Prop. IFX: 0.210 (13%)	*Full model code*	YES
0.404 (9.9%)		3.18 (9.1%)	1.64 (6.3%)		0.344 (20%)	ALB~CL: −1.13 (36%)	V1: 22.7% (16%)	Prop. TNF: 0.406 (9%)	*ODEs/model schematic*	YES
TMDD model							V2: 74.2% (19%)		*Covariate relationships*	YES
*B_maxTNF_ (pM)*		*K_ss,TNF-IFX_ (nM)*	*K_e(p),TNF-IFX_ (1/d)*		*K_deg,TNF (_1/d)*		CL-V1: 12.3% (106%)		*Random effects*	YES
0.38 (20%)		14 (24%)	0.984 (19%)		5.12 FIX		B_max,TNF_: 39.4% (16%)			
**Dreesen (2019)** [56]: 1-CMT model, first order elimination										
*k_e_ (1/d)*		*V (L)*				CRP~k_e_: 0.0859 (25%)	k_e_: 29.8% (7.8%)	Prop.: 19.2% (9.4%)	*Full model code*	YES
MAYO 1: 0.0521 (11%)		6.34 (3.2%)				ALB~k_e_: −0.808 (39%)	V: 26.5% (20%)	Add.: 0.300 ug/mL (FIX)	*ODEs/model schematic*	YES
MAYO 2: 0.0543 (3.6%)						FFM~V: 0.544 (29%)	IOV k_e_: 18.7% (17%)		*Covariate relationships*	YES
MAYO 3: 0.0667 (8.1%)						CORT~V: 1.33 (6.9%)			*Random effects*	YES
						Pancolitis~V: 1.23 (0.019%)				
**Hanzel (2021)** [41]: 2-CMT model, first order absorption (s.c. administration), first order elimination										
*CL (L/d)*	*V1 (L)*	*V2 (L)*	*Q (L/d)*	*k_a_ (1/d)*	*F(%)*	ALB~CL: −0.826 (11%)	CL: 27.7% (14%)	Prop.: 0.102 (2%)	*Full model code*	NO
0.355 (2%)	3.10 (3%)	1.93 (2%)	0.598 (4%)	0.273 (8%)	79.1 (3%)	BW~CL: 0.666 (15%)	V1: 21.4% (21%)	Add.: 1.66 (3%) ug/mL	*ODEs/model schematic*	YES
						ADA_cat_~CL: 1,39 (3%)	k_a_ 48.5% (45%)		*Covariate relationships*	YES
						BW~V1: 0.385 (34%)	F: 16.4% (15%)		*Random effects*	YES
						BW~V2: 1.08 (9%)	CL-V1:0.028% (31.4%)			
						BW~Q: 1.26 (15%)	CL-k_a_: −0.046% (53.2%)			
							CL-F1: −0.013% (42.2%)			
							V1-ka: −0.069% (47%)			
							V1-F: 0.00008% (744%)			
							k_a_-F: 0.003% (413%)			
							IOV CL: 17.5% (6%)			
**Grisic (2021)** [59]: 2-CMT model, first order elimination									NO	YES
*CL (L/h)*		*V1 (L)*		*V2 (L)*	*Q (L/h)*					
0.0109 (3%)		3.67 (-)		0.956 (11%)	0.0067 (-)	ALB~CL: −1.17 (21%)	CL: 34.9% (8%)	Prop.: 24% (14%)	*Full model code*	NO
						ADA_cat_~CL: 0.972 (4%)	V1: 12.8% (-)	Add.(SD): 0.478 (21%) µg/mL	*ODEs/model schematic*	YES
						IMM~CL: 0.847 (5%)	V2: 55.3% (-)		*Covariate relationships*	YES
						WT~CL: 0.356 (41%)			*Random effects*	YES

ADA: Anti-drug antibodies; ADA_cat_: Categorical ADA; Add: additional residual variability; ALB: Albumin concentrations; AZA: Azathioprine; B_maxTNF_: Baseline TNF concentration; BW: Body weight; CD: Crohn’s disease; CL: Clearance; CL_base_: Baseline clearance; CL_time_: Time dependant CL; CMT: Compartment; CORT: Corticosteroids; CRP: C-Reactive protein; F: Bioavailability; FFM: Free fat mass; HBI: Harvey–Bradshaw index; IFX: Infliximab; IIV: Interindividual variability; IMM: Immunomodulators; IOV: Interoccasion variability; ka: Absorption rate constant; K_ss,TNF-IFX_: Steady-state equilibrium constant; k_deg,TNF_: Degradation constant TNF receptor; ke: elimination rate constant; k_e(p),TNF-IFX_: Internalisation rate complex; MTX: Methotrexate; NR: Not reported; ODE: Ordinary differential equation; Prop: Proportional residual variability; Q: Intercompartmental clearance; V1: Central volume of distribution; V2: Peripheral volume of distribution; RA: Rheumatoid arthritis; s.c.: subcutaneous; SD: Standard deviation; TMDD: Target-mediated drug disposition; TNF: Tumour necrosis factor; UC: Ulcerative colitis; WT: Weight; y.o.: Years old.

**Table 3 pharmaceutics-14-02095-t003:** Reported model performance and evaluation of the published models and their applications.

Model Article	Reported Goodness of Fit (GOF) Evaluation	Reported GOF Performance	Reported Advanced model Evaluation (AME)	Reported AME Performance	Application Model	Fit for Purpose?
Ternant-model (2008)						
Ternant (2008) [48]	OBS vs. IPRED on normal scale WRES vs. IPRED Individual fits of some representative patients	Acceptable performance but lack of PRED, underestimation of ADA-CL as seen at very low concentrations.	NR	NA	Descriptive Covariate analysis	Yes (no predictions)
Fasanmade-model (2009)						
Fasanmade (2009) [49]	OBS vs. IPRED and OBS vs. PRED on normal scale CWRES vs. PRED and time Individual fits of some representative patients	Acceptable performance	Bootstrap	Acceptable but lack of predictive performance evaluation	Descriptive Covariate analysis	Yes (no predictions)
Vande Casteele (2019) [46]	NR	NA	NR	NA	Descriptive Link to PD Evaluate association individual CL and remission	Yes (same data as original model, no PK predictions)
Kimura (2019) [47]	NR	NA	NR	NA	Descriptive Link to PD Link to outcome	No (PK was predicted for a new population without any evaluation of predictive performance)
Kimura (2020) [43]	OBS vs. IPRED RES vs. time	Acceptable performance but lack of PRED, (C)WRES	NR	NA	Descriptive Link to PD Link to outcome	No (PK was predicted for a new population without any evaluation of predictive performance)
Schräpel (2021) [61]	OBS vs. IPRED	Acceptable except for lower concentrations	Accuracy (ζ) and bias (SSPB) pvcVPC	ADA_−_: Moderate bias and imprecision, variability slightly overestimated, ADA_+_: poor performance	External evaluation	Yes
Fasanmade-model (2011, CD)						
Fasanmade (2011) [64]	OBS vs. IPRED and OBS vs. PRED on normal scale CWRES vs. PRED and time Individual fits of some representative patients	Acceptable performance	Bootstrap VPC	Acceptable but VPC difficult to interpret due to lack of observed percentiles	Descriptive Covariate analysis	Yes (no predictions)
Santacana (2018) [37]	OBS vs. IPRED	Acceptable performance	Relative and absolute median individual pre-diction error pcVPC NPDE	Low bias and imprecision, good predictive performance not supported	External evaluation	Yes
Kimura (2019) [47]	NR	NA	NR	NA	Descriptive Link to PD Link to outcome	No (New population without any evaluation of predictive performance)
Kimura (2020) [43]	OBS vs. IPRED RES vs. time	Acceptable performance but lack of PRED, (C)WRES	NR	NA	Descriptive Link to PD Link to outcome	No (New population without any evaluation of predictive performance)
Gil Candel (2020) [40]	OBS vs. IPRED and OBS vs. PRED	Acceptable performance	Mean (absolute) prediction error	Acceptable performance	External evaluation	Yes
Santacana (2020) [39]	Santacana 2018	Santacana 2018	Santacana 2018	Santacana 2018	Bayesian forecasting and updating	Yes (Model was evaluated by the authors in previous publication)
Bauman (2020) [58]	OBS vs. IPRED and OBS vs. PRED on log-scale CWRES vs. PRED and time	Slight bias: underprediction lowest concentrations	NR	NA	Bayesian forecasting and updating	No (Very basic model evaluation and model used for predictions)
Schräpel (2021) [61]	OBS vs. IPRED	Acceptable except for lower concentrations	Accuracy (ζ) and bias (SSPB) pvcVPC	ADA_−_: low bias and imprecision, good predictive performance, ADA_+_: poor performance	External evaluation	Yes
Xu-model (2012) [65]						
Dubinsky (2017) [53]	NR	NA	NR	NA	Bayesian forecasting and updating	No (New population without any evaluation of predictive performance)
Wojciechowski (2017) [44]	NR	NA	NR	NA	Bayesian forecasting and updating	No (New population without any evaluation of predictive performance)
Eser (2018) [54]	NR	NA	Bias estimation depending on the number of samples used for PK estimation (RMSE)	NA	Comparison of assay influence and amount of observations needed for Bayesian forecasting	Yes (but more evaluation needed before predicting)
Schräpel (2021) [61]	OBS vs. IPRED	Acceptable	Accuracy (ζ) and bias (SSPB) pvcVPC	ADA_−_: Moderate bias and imprecision, variability slightly overestimated, ADA_+_: poor performance	External evaluation	Yes
Dubinsky (2022) [42]	NR	NA	NR	NA	Bayesian forecasting and updating	No (New population without any evaluation of predictive performance)
Dotan-model (2014)						
Dotan (2014) [36]	Individual plots with PRED, IPRED and DV vs. TIME	NA	VPC mentioned but NR	NA	Descriptive Covariate analysis Simulations Dosing optimization	No (Application of model for dosing simulations without thorough evaluation)
Buurman-model (2015)						
Buurman (2015) [50]	Individual plots with PRED, IPRED and DV vs. TIME for 9 representative patients GOF plots mentioned by the authors but NR	NA	VPC	Acceptable	Descriptive Covariate analysis Simulations Dosing optimization	Yes (If model evaluation is as good as said (not shown))
Schräpel (2021) [61]	OBS vs. IPRED	Acceptable	Accuracy (ζ) and bias (SSPB) pvcVPC	ADA_−_: Moderate bias and imprecision, good predictive performance, ADA_+_: poor performance	External evaluation	Yes
Passot-model (2016)						
Passot (2016) [51]	OBS vs. IPRED and OBS vs. PRED on normal scale CWRES vs. PRED and time Individual fits of some representative patients	Acceptable	NPDE	Acceptable	Descriptive Covariate analysis	Yes
Schräpel (2021) [61]	OBS vs. IPRED	Acceptable except for lower concentrations	Accuracy (ζ) and bias (SSPB) pvcVPC	ADA_−_: Low bias and imprecision, good predictive performance, ADA_+_: poor performance	External evaluation	Yes
Brandse-model (2017)						
Brandse (2017) [52]	NR	NA	VPC	Slight underprediction lower percentile	Descriptive Link to ADA formation	Yes (no predictions)
Schräpel (2021) [61]	OBS vs. IPRED	Acceptable except for lower concentrations	Accuracy (ζ) and bias (SSPB) pvcVPC	ADA_−_: Moderate bias and imprecision, variability slightly overestimated, ADA_+_: poor performance	External evaluation	Yes
Kevans-model (2018)						
Kevans (2018) [55]	GOF plots mentioned by the authors but NR	NA	AME plots mentioned by the authors but NR	NA	Descriptive Covariate analysis Link to outcome	Yes (If model evaluation is as good as said (not shown))
Petitcollin-model (2019)						
Petitcollin (2019) [38]	OBS vs. IPRED and OBS vs. PRED on normal scale IWRES vs. PRED 4 representative individual fits	Acceptable	VPC	Slight underprediction lower percentile	Descriptive Covariate analysis Link to outcome	Yes
Schräpel (2021) [61]	OBS vs. IPRED	Acceptable except for lower concentrations	Accuracy (ζ) and bias (SSPB) pvcVPC	ADA_−_: Low bias and imprecision, variability largely overestimated, ADA_+_: poor performance	External evaluation	Yes
Berends-model (2019)						
Berends (2019) [57]	OBS vs. IPRED and OBS vs. PRED on normal scale CWRES vs. PRED	Acceptable	Bootstrap VPC	Acceptable	Descriptive Mechanistic insights	Yes
Dreesen-model (2019)						
Dreesen (2019) [56]	OBS vs. IPRED and OBS vs. PRED on normal scale CWRES vs. PRED and time	Acceptable	pcVPC Bootstrap	Slight underprediction lower percentile	Descriptive Covariate analysis Link to outcome	Yes
Faelens (2021) [45]	NR	NA	pcVPC	Slight underprediction lower percentile	Dosing evaluation	Assumption dependant
Hanzel-model (2021)						
Hanzel (2021) [41]	GOF plots mentioned by the authors but NR	NA	Bootstrap pcVPC	Acceptable	Descriptive Covariate analysis Dosing evaluation	Yes
Grisic-model (2021)						
Grisic (2021) [59]	NR	NA	pcVPC	Acceptable	Descriptive Covariate analysis Link to PD Dosing evaluation	

ADA: Antidrug antibody; AME: Advanced model evaluation; CL: Clearance; CWRES: Conditional weighted RES; DV: Dependent variable; GOF: Goodness of fit; IPRED: Individual predictions; IWRES: Individual weighted RES: NA: Not applicable; NPDE: Normalised probability distribution error; NR: Not reported; OBS: Observations; pcVPC; prediction-corrected VPC; pvcVPC: prediction- and variability-corrected VPC; PD: Pharmacodynamics; PK: Pharmacokinetics; PRED: Population predictions; RES: Residuals; SSPB: Symmetric signed percentage bias; VPC: Visual predictive check; ζ: Median symmetric accuracy.

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
