# Peer review of "Ulcerative Colitis and Acute Severe Ulcerative Colitis Patients Are Overlooked in Infliximab Population Pharmacokinetic Models: Results from a Comprehensive Review"

_pharmaceutics, 2022, doi:10.3390/pharmaceutics14102095_

Round 1

Reviewer 1 Report

The authors prepared a review manuscript regarding “Ulcerative colitis and acute severe ulcerative colitis patients are 2 overlooked in infliximab population pharmacokinetic models:  results from a comprehensive review”. This is an interesting review article described the PopPK models for IFX from published literature. This manuscript requires some minor modifications before acceptance.

Please see my comments below and include these modifications in the manuscript.

Figure 1 is not clear and needs to be replaced.

The authors can include the figure showing the pharmacological mechanism of IFX for IBD, UC, CD.

The authors should include the figure showing the effect of population variables on pharmacokinetic properties of IFX.

Reviewer 2 Report

In this paper, Alix and co-authors have systemically analyzed published modelling papers related to infliximab PK in IBD patients and accessed the applicability of PK models in clinical setting. They concluded that PK modeling work on UC patients is very limited and disease activity is poorly characterized.

This review study is very interesting and meaningful, which provides crucial directions for future studies on IFX PK modeling. However, some information is missing, and I have two suggestions and comments which I think should be addressed before being published.

1. More basic introduction is required for clinical treatment of IBD patients, specifically those with UC. Current review mentioned acute colectomy, corticosteroid and IFX, but didn’t specify on what circumstances each intervention is warranted. For example, the authors didn’t mention that IFX could be used to eliminate corticosteroid use in adult patients with moderately to severely active disease who have had an inadequate response to conventional therapy. Moreover, the authors didn’t mention if there’re other drug available for treating UC patients

2. The authors mentioned that UC is more prevalent than CD in page 1 line 42, but patient population in the 26 evaluated publications suggest otherwise. Majority of the 26 PK modeling studies were conducted on CD patients, which could be due to the fact IFX is not the only treatment for UC patients.
